# Microbial Composition and Stool Short Chain Fatty Acid Levels in Fibromyalgia

**DOI:** 10.3390/ijerph20043183

**Published:** 2023-02-11

**Authors:** Yunkyung Kim, Geun-Tae Kim, Jihun Kang

**Affiliations:** 1Division of Rheumatology, Department of Internal Medicine, Kosin University College of Medicine, Busan 49267, Republic of Korea; 2Department of Family Medicine, Kosin University Gospel Hospital, Kosin University College of Medicine, Busan 49267, Republic of Korea

**Keywords:** fibromyalgia, gastrointestinal microbiome, short chain fatty acids, metabolome, brain–gut axis

## Abstract

Background: The present study aimed to evaluate microbial diversity, taxonomic profiles, and fecal short chain fatty acid (SCFA) in female patients with fibromyalgia syndrome (FMS). Methods: Forty participants (19 patients with FMS and 21 controls) were included in the study, and the diagnosis of FMS was made based on the revised American College of Rheumatology criteria. DNA extraction from fecal samples and 16S rRNA gene sequencing were conducted to estimate microbial composition. To compare alpha diversity, the Shannon index accounting for both evenness and richness, Pielou’s evenness, and Faith’s phylogenetic diversity (PD) were calculated. Unweighted and weighted UniFrac distances, Jaccard distance, and Bray–Curtis dissimilarity were used to calculate beta diversity. Furthermore, stool metabolites were analyzed using gas chromatography-mass spectrometry, and a generalized regression model was used to compare the SCFA of stools between FMS and healthy controls. Results: Compared with the control, patients with FMS had lower observed OTU (*p* = 0.048), Shannon’s index (*p* = 0.044), and evenness (*p* < 0.001). Although patients with FMS had a lower PD than did controls, statistical significance was not reached. We observed significant differences in unweighted (*p* = 0.007), weighted UniFrac-based diversity (*p* < 0.005), Jaccard distance (*p* < 0.001), and Bray–Curtis dissimilarity (*p* < 0.001) between the two groups. Although the FMS groups showed lower propionate levels compared with those of the control group, only marginal significance was observed (0.82 [0.051] mg/g in FMS vs. 1.16 [0.077] mg/g in the control group, *p* = 0.069). Conclusions: The diversity of the microbiome in the FMS group was lower than that in the control group, and the reduced stool propionate levels could be associated with the decreased abundance of propionate-producing bacteria.

## 1. Introduction

Fibromyalgia syndrome (FMS) is characterized by widespread chronic pain accompanied by fatigue, depression, sleep disturbance, and cognitive impairment [1]. The global mean prevalence of FMS is 2.7%; however, it varies from 0.4% to 9.3% across countries, and is more prevalent in women [2]. Although the mechanism of pain in the central nervous system (CNS) has been proposed as a plausible hypothesis for FMS, the pathogenesis of this chronic condition is largely unknown. An accumulating body of evidence has highlighted the existence of bidirectional interplay between the microbiome and CNS, and this interaction [3], the so-called gut–brain axis, has an impact on mood change [3,4,5], sleep disorders [6] and cognitive impairment [7], which are commonly reported in patients with FMS. In addition, recent studies have revealed that the composition of the gut microbiome can play a role in the central sensitization of chronic pain by regulating astrocytes, microglia, and other immune cells [8]. 

Although the growing evidence on the gut–brain axis makes the potential role of the microbiome in the pathogenesis of FMS an area of research interest, previous studies on the association between the composition of the microbiome and FMS have revealed conflicting findings. Two studies in Canada (77 patients with FMS and 79 controls) [9] and Spain (105 patients with FMS and 54 controls) [10] showed that the diversity of the gut microbiome was reduced in patients with FMS compared to that in participants without FMS. Notably, a Canadian study explored the impact of changes in microbiome composition on serum metabolites, revealing reduced serum butyrate and propionate levels in participants with FMS [9]. In addition, a Spanish study measured serum metabolite levels, reporting that glutamate and serine are involved in neurotransmitter metabolism [10]. However, another study from Austria that included 25 patients with FMS and 26 age- and sex-matched controls failed to show any significant differences in gut bacterial diversity between the two groups [11]. 

However, because most previous studies have been conducted among Caucasian populations, the gut microbiome of patients with FMS in the Asian population, which has distinctive dietary patterns from the Western population, has not been evaluated [12]. Consequently, the composition and diversity of the microbiome in FMS of non-Caucasian ethnicities are largely unknown. In addition, the levels of metabolites, including short chain fatty acids (SCFA), were measured in serum, and there is a knowledge gap regarding the association between changes in bacterial composition and fecal SCFA levels in patients with FMS. Therefore, the present study aimed to evaluate the microbial diversity, taxonomic profiles, and fecal SCFA in Korean women with FMS. 

## 2. Materials and Methods

### 2.1. Study Participants

This study included participants aged ≥19 years who visited the rheumatology outpatient clinic of a tertiary hospital between January 2021 and December 2021. Individuals who were administered antibiotics and experienced acute infection in the gastrointestinal tract within 4 weeks were not included in the study. Nineteen patients with FMS participated in the study, and the diagnosis of FMS was made based on the revised American College of Rheumatology criteria [1]. For the control group, 21 healthy controls who participated in annual health checkups at the Department of Disease Prevention and Health Promotion without evidence of inflammatory rheumatologic diseases were enrolled in the study. The same exclusion criteria were used for the control group. Ultimately, 40 participants were included in the analysis. The study protocol complied with the principles of the Declaration of Helsinki, and written informed consent was obtained from all the participants. This study was reviewed and approved by the Institutional Review Board of the Kosin University Medical School (IRB file No. KUGH-2020-05-023).

### 2.2. Data Collection and Measurements

Information regarding anthropometric measurements, health behaviors, and symptomatic scales of the FMS was obtained through face-to-face interviews with medical personnel. To verify the data on anthropometric measurements (height and weight) and health behaviors (smoking status), electronic medical records were used. Smoking status (smokers and non-smokers) and alcohol consumption (yes or no) were categorized into two groups. Body mass index (BMI) was measured in kg/m^2^. We used the participants’ medical records to gather information on comorbidities, such as hypertension, diabetes, and dyslipidemia. Hypertension was defined as SBP ≥ 140 mmHg, DBP ≥ 90 mmHg, or the use of antihypertensive medication. Diabetes was defined as a fasting glucose level of ≥ 100 mg/dL or the use of anti-diabetic medications. Participants with a low-density lipoprotein (LDL) cholesterol level ≥ 130 mg/dL, high-density lipoprotein (HDL) cholesterol level < 40 mg/dL, triglyceride level ≥ 150 mg, or who used lipid-lowering medications were defined as having dyslipidemia.

Venous blood was obtained after at least 8 h of fasting, and biochemical analysis was conducted at a diagnostic laboratory where an external quality assessment program was applied on a regular basis. The levels of total cholesterol, serum triglyceride, HDL and LDL cholesterol, and fasting glucose were measured using an enzymatic method, a two-reagent homogenous method, and the hexokinase G-6-PDH method, respectively (au 5800 Analyzer, Beckman Coulter, Brea, CA, USA). To measure high-sensitivity C-reactive protein (hs-CRP) and erythrocyte sedimentation rate (ESR), the immunoturbidimetry method (Cobas8000, Roche, Mannheim, Germany) and an automated analyzer with photometric capillary stopped flow kinetic analysis (Alifax SpA, Polverara, Italy) were used. A self-reported questionnaire was used to assess the widespread pain index (WPI), symptom severity scale (SS), visual analog scale (VAS) for pain, and fibromyalgia impact questionnaire (FIQ) [13].

### 2.3. DNA Extraction from Fecal Samples and 16S rRNA Gene Sequencing

Fecal samples were collected using a kit designed to store fecal specimens at room temperature. The accuracy and reliability of the kit have been validated in a previous study [14]. DNA was extracted from fecal specimens using the MOBio PowerSoil DNA Isolation Kit (MO BIO Laboratories, Carlsbad, CA, USA) within 4 weeks of collection. To target and amplify the V3 and V4 regions of the 16S rRNA, the universal primers rRNA. 

(Forward: TCGTCGGCAGCGTCAGATGTGTATAAGAGACAGCCTACGGGNGGCWGCAG, and Reverse: GTCTCGTGGGCTCGGAGATGTGTATAAGAGACAGGACTACHVGGGTATCTAATCC) were used with a combination of indexing barcodes (Nextera XT DNA, Library Preparation kit (Illumina, San Diego, CA, USA). Pooled stool samples were sequenced on an Illumina MiSeq platform (Illumina, San Diego, CA, USA) according to the manufacturer’s instructions [15,16]. DADA2 plugged into the QIIME2 package (version 2022.6, https://qiime2.org, accessed on 5 August 2022) was used to conduct sequence quality control, filtering low quality sequences and removing chimeras, and to generate amplicon sequence variants (ASVs) regarded as 100% operational taxonomic units [17]. For taxonomic analysis, a pre-trained naïve Bayes classifier and the q2-feature-classifier against the Greengene 99% OTUs (version 13_8) of the 16S rRNA gene were used to assign taxonomy to ASVs. 

### 2.4. Measurement of SCFA in Fecal Samples

SCFA were extracted from 0.2 g of fecal samples. Fecal samples were immediately frozen at −20 °C and transferred to a −70 °C freezer without preservatives. Fecal matter was first homogenized in three volumes of deionized water, centrifuged for 3 min at 13,000 rpm, and, finally, the supernatant was collected. The supernatant (150 μL) was placed in a 10-mL screw cap vial with 150 µL GC buffer solution. A solution containing (NH_4_)_2_SO_4_, NaH_2_PO_4_, and 2-ethylbutric acid was used as an internal standard. Stool SCFA were analyzed using a gas chromatography-mass spectrometry (GC-MS) system (7890B, Agilent Technologies, Santa Clara, CA, USA) equipped with a 7697A headspace sampler and flame ionization detector (FID) (Agilent Technologies). An HP-innowax capillary GC column (30 m × 0.32 mm × 0.25 μm; Agilent) was used with a constant flow of nitrogen as a carrier gas. The operating conditions were as follows: oven temperature: 85 °C, loop temperature: 90 °C, transfer line temperature: 100 °C, and FID temperature: 250 °C. The column temperature was raised from 60 °C to 140 °C at 30 °C per minute, then increased to 170 °C at 30 °C per minute, and finally to 180 °C at 40 °C per minute and held for 0.75 min. The homogeneity of the chromatographic peaks was verified using the extracted ions of characteristic fragments to optimize resolution and peak symmetry. Data analysis was performed using MassHunter WorkStation (Agilent Technologies). Concentrations of SCFA were expressed as μmol/g feces.

### 2.5. Statistical Analysis 

The general characteristics of the study participants were compared between the FMS and control groups, using the *t*-test for continuous variables and the chi-square test for categorical variables. 

The number of ASVs observed in each sample, Shannon index accounting for both evenness and richness, Pielou’s evenness, and Faith’s phylogenetic diversity (PD) were estimated to compare the alpha diversity. The Kruskal–Wallis test was performed to compare pairwise differences in the non-parametric variables. We calculated beta diversity to estimate dissimilarity among group members using the UniFrac distance, accounting for the phylogenetic distances between ASVs. Unweighted and weighted UniFrac distances were used to incorporate the presence/absence and abundance of ASVs, respectively, into the analysis models. Bray–Curtis dissimilarities, a non-phylogenic index, were calculated for the abundance data. To test the significance of the differences between groups, pairwise permutational multivariate analysis of variance (PERMANOVA) with 999 random permutations was conducted [18]. The diversity of the microbiome is presented using the box and principle of the component plots. The abundance of microbiomes between participants with and without FMS was compared using analysis of compositions of microbiomes with bias correction (ANCOM-BC), designed to correct bias that might occur in the analysis of microbial composition [19].

A generalized regression model was used to compare SCFA of stool between patients with FMS and healthy controls, adjusting for age and BMI. We presented the differences in stool acetate, propionate, isobutyrate, butyrate, isovalerate, and valerate between the two groups using boxplots. In addition, we conducted a subgroup analysis to test whether inflammatory markers and symptomatic scales of FMS, such as WPI, SS, VAS, and FIQ, were associated with alpha diversity of the microbial composition. Spearman’s correlation analysis was used, and correlations among variables were presented using a heatmap. Two-tailed tests were performed in all analyses, and *p*-values < 0.05 were considered statistically significant. All analyses were performed using IBM SPSS Statistics for Windows version 24.0 (IBM Corp., Armonk, NY, USA) and QIIME 2 (version 2022.6, https://qiime2.org) [20].

## 3. Results

### 3.1. General Characteristics of the Study Participants

The general characteristics of the participants are listed in Table 1. The mean ages of the FMS and control group were 51.4 (7.4) years, and 46.6 (8.7) years, respectively. The FMS group had a higher mean BMI compared with that of the control group (27.8 ± 4.8 kg/m^2^ vs. 22.1 ± 2.7 kg/m^2^, Pp < 0.001). While the proportion of current smokers and dyslipidemia was higher in participants with FMS, there was no difference in inflammatory markers, including CRP and ESR. 

### 3.2. Gut Microbial Diversity within and between FMS and the Control Groups

The alpha diversity of the gut microbial taxa between participants with and without FMS is shown in Figure 1. Compared with the control, FMS showed lower in observed OTU (*p* = 0.048), Shannon’s index (*p* = 0.044), and evenness (*p* < 0.001). Although FMS had a lower PD than the control, statistical significance was not reached. 

There were significant differences in both non-phylogenetic (Jaccard distance and Bray–Curtis dissimilarity) and phylogenetic (unweighted and weighted UniFrac-based diversity) diversities between the FMS and non-FMS groups (Figure 2). We observed significant differences in the unweighted (*p* = 0.007), weighted UniFrac-based diversity (*p* < 0.005), Jaccard distance (*p* < 0.001), and Bray–Curtis dissimilarity (*p* < 0.001) between the two groups. However, distinctively separated patterns by principal coordinate analysis were not found because of interindividual variation with a relatively small sample size. 

### 3.3. Abundance of Microbial Composition and FMS 

Among the 187 identified genera, 46 genera were significantly different in abundance between the FMS and control groups (Table 2), while *Frisingicoccus*, *Caproiciproducens*, *Eisenbergiella*, *Catenibacillus*, *Paludicola*, *Megasphaera*, *Howardella*, *Eubacterium fissicatena group*, *Clostridia*, *Victivallis*, *Slackia*, *Lachnospiraceae_NC2004_group*, *Succinivibrio*, *Coprobacillus*, *Faecalitalea*, *Tuzzerella*, *Fournierella*, *Actinomyces*, *Staphylococcus*, *Fenollaria*, *Corynebacterium*, *Ornithobacterium*, *Porphyromonas*, and *Peptoniphilus* were more abundant in the FMS group compared with the non-FMS group. *Prevotellaceae UCG-001*, *Parvimonas*, *Campylobacter*, *Finegoldia*, *Gemella*, *Terrisporobacter*, *Granulicatella*, *Methylobacterium-Methylorubrum*, *Gastranaerophilales*, *Allisonella*, *Enterorhabdus*, *Butyricicoccaceae*, *Peptococcus*, *Anaerococcus*, *Methanobrevibacter*, *Adlercreutzia*, *Coprobacter*, *Rikenellaceae RC9 gut group*, *Acinetobacter*, *Alloprevotella*, *Eubacterium ruminantium group*, and *Eubacterium eligens group* were less abundant in the FMS group. 

### 3.4. Association of Stool SCFA and FMS

The levels of stool acetate, propionate, isobutyrate, butyrate, isovalerate, and valerate are shown in Figure 3. Although the FMS groups showed lower propionate levels than the control group, only marginal significance was observed (0.82 [0.051] mg/g in FMS vs. 1.16 [0.077] mg/g in the control, *p* = 0.069). There were no significant differences in other stool SCFA levels between the two groups. 

### 3.5. Correlation between Inflammatory Markers, Symptomatic Scales of FMS and Microbial Diversity and SCFA

General correlations among the symptomatic scales of FMS, serum inflammatory markers as clinical variables, microbial diversity, and fecal SCFA levels are presented as Spearman’s correlation heatmaps (Figure 4). In the subgroup analysis, symptomatic scales of FMS, such as the VAS, WPI, and FIQ were not significantly correlated with the observed ASVs and Simpson and Shannon indices, which indicate microbial diversity. In addition, they were not significantly correlated with SCFA levels. Serum inflammatory markers were not significantly correlated with microbial diversity or fecal metabolites.

## 4. Discussion

This study is the first to evaluate the association between FMS and gut bacterial composition and abundance in Korean women. Patients with FMS had lower phylogenetic and non-phylogenetic measures of alpha diversity for the microbiota than did participants in the control group. Although stool propionate levels tended to decrease in the FMS group compared with the control group, only marginal significance was found. Serum inflammatory markers and symptomatic scales of FMS did not correlate with microbial diversity or SCFA.

Consistent with the present study, previous studies in Canada [9] and in the US [10] reported that patients with FMS had lower gut microbial diversity compared to non-FMS participants. Although it is unclear how decreased microbial diversity contributes to the pathogenesis of FMS, decreased diversity of the gut microbial community could lead to intestinal dysbiosis, which is associated with increased permeability of the intestine [21]. Elevated intestinal permeability is related to the inflammatory reaction of the bowels [22], which might alter the manner in which sensory neurons respond to pain. A previous study on altered intestinal permeability among patients with FMS also supports this hypothesis.

With respect to the abundance of genera, *Eisenbergiella* and *Coprobacillus* were elevated in the FMS group, which was consistent with previous studies. Other genera that were more abundant in the FMS group than in the control group were associated with the production of SCFA and amino acids. *Caproiciproducens* is an anaerobic bacterium that produces acetate, butyrate, and caproate [23], and Clostridium is associated with the production of propionate, which is involved in the sensitization of pain receptors [24,25]. *Succinivibrio* and *Coprobacillus* are relevant to tryptophan, which is involved in strengthening tight junctions [26] and has an influence on the brain–gut axis via the interaction between tryptophan metabolites and microglia and astrocytes [27]. In addition, *Frisingicoccus* and *Enterobacter* are associated with Parkinson’s disease [28,29], and victivallis are increased in patients with stroke [30]. 

Several bacteria that were less abundant in patients with FMS than in participants in the control group were associated with symptoms of FMS in addition to pain. A decreased abundance of *Eubacterium ruminantium* has been reported to be associated with anxiety and depressive-like behavior [31]. A previous study revealed that the abundance of *Methanobrevibacter* was lower in patients with migraine than in participants without migraine [32]. The association of *Enterorhabdus* with inflammatory bowel disease was revealed in an experimental study [33], and this finding, at least in part, supports the association between the gut inflammatory response and the composition of gut microbiota. 

Among stool SCFA, propionate levels were lower in the FMS group than in the control group, with marginal significance. In addition, a previous study in Canada indicated reduced levels of serum propionate in FMS, at least in part, which is also similar to our findings [9]. Previous studies revealed that propionate plays a protective role in atopy [34] and bronchial asthma [35], and might alleviate rheumatoid arthritis by inhibiting the proliferation of fibroblast-like synoviocytes [36]. SCFA has also been implicated in the development of Alzheimer’s disease [37] and could exert beneficial effects on motor symptoms of Parkinson’s disease [38], inferring the influence of microbiome-derived propionate on the brain–gut axis. A recent study indicated that propionate is involved in the regeneration and functional recovery of sensory axons and supports the role of propionate in the perception and processing of sensory pain via the brain–gut axis [39]. However, considering the limited literature evidence and relatively small sample size, further prospective studies are warranted to evaluate serial changes in stool propionate and the risk of FMS. 

The abundance of several propionate-producing bacteria, including *Eubacterium* and *Provotella*, was reduced in the FMS group compared to the non-FMS group, and this finding was consistent with a previous study of microbiota for FMS. In particular, *Eubacterium* sp. *Eligens*, an SCFA-producing bacterium with anti-inflammatory ability [40], was significantly reduced in patients with FMS. Moreover, the reduced abundance of *Prevotella*, a propionate-producing bacterium associated with abdominal pain in the general population [41], also infers the potential role of bacteria-derived propionate in the pathogenesis of FMS. However, a few studies have reported an increased abundance of *Prevotella* in individuals with major depressive mood, at least in part, arguing against the association of *Prevotella* and FMS [4,5].

The present study has several limitations. First, the number of participants was relatively small, and caution should be exercised when interpreting the study results. However, our study findings were largely in line with those of previous studies reporting reduced diversity of the microbiome in the FMS groups compared with those of controls. Furthermore, changes in the abundance of *Eubacterium*, *Eisenbergiella*, and *Coprobacillus* strengthen the study’s findings on the relationship between the composition of the microbiome and FMS. Second, because we could not assess the dietary factors that had a significant impact on the microbial composition of the gut, residual confounding effects related to the diet pattern might have an influence on the bacterial composition of the gut. Third, the study participants were solely composed of Koreans; therefore, the study findings may not be generalizable to other ethnicities with different dietary patterns and environmental exposures. Fourth, as there were no smokers in the control group, the negative effect of smoking on the gut microbiome could not be assessed. Fifth, information on diet was not included, and the effect of dietary habits on the composition of the microbiota in the FMS group was not evaluated. Despite these limitations, the present study extended the knowledge of the composition of the gut microbiome in Korean patients with FMS and revealed a potential association between FMS and microbial metabolites, using stool metabolite analysis.

## 5. Conclusions

In conclusion, the diversity of the microbiome in FMS was lower than that in the control group, and a distinctive pattern of the taxonomic profile of FMS was found. In addition, the stool propionate level, which is involved in the sensitization of pain reception, was lower in the FMS group than in the non-FMS group; moreover, the reduced propionate level could be associated with the decreased abundance of propionate-producing bacteria, such as *Eubacterium* and *Prevotella*. Further large-scale studies are warranted to replicate and confirm the pathognomonic gut microbiome in FMS patients.

## Figures and Tables

**Figure 1 ijerph-20-03183-f001:**
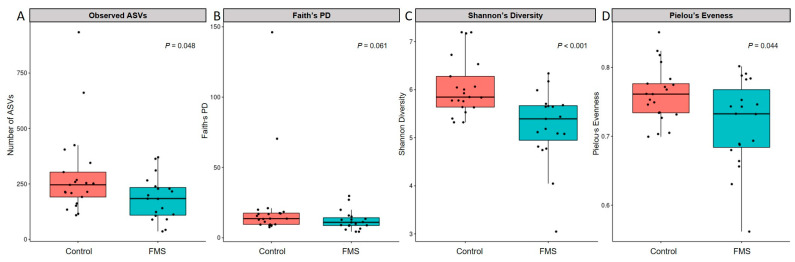
Alpha diversity indices for each sample group. (**A**). Observed ASVs, (**B**). Faith’s phylogenetic diversity (PD), (**C**). Shannon’s Diversity, (**D**). Pielou’s Evenness. Dots indicate the diversity value of each fecal sample.

**Figure 2 ijerph-20-03183-f002:**
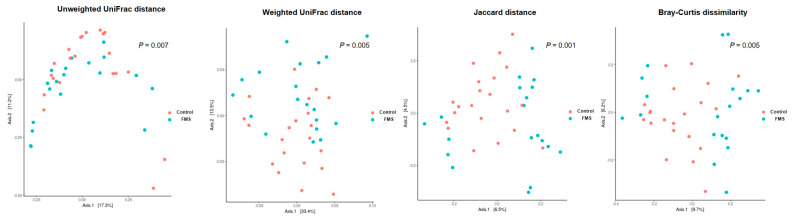
Principal component Analysis (PCoA) of unweighted UniFrac distance, weighted UniFrac distance, Jaccard distance, and Bray−Curtis dissimilarity of the OTU abundance between control and FMS group.

**Figure 3 ijerph-20-03183-f003:**
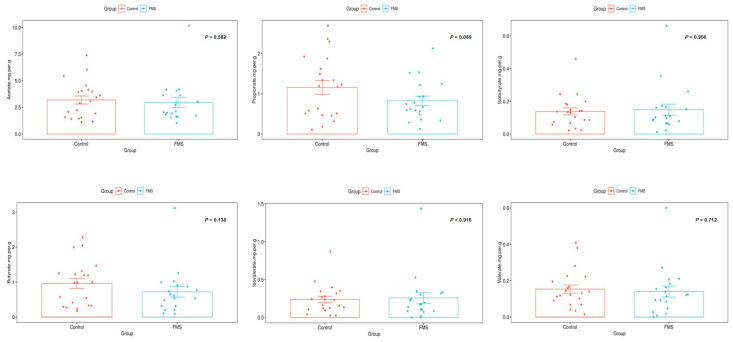
The levels of stool acetate, propionate, isobutyrate, butyrate, isovalerate, and valerate between control and FMS groups.

**Figure 4 ijerph-20-03183-f004:**
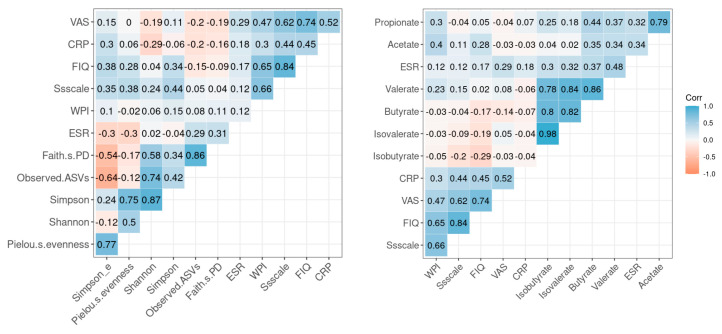
Heat map of Spearman correlation matrices for inflammatory markers, symptomatic scales of FMS, microbial diversity, and fecal metabolites. Color gradients indicate degree of correlation; positive correlations are marked in blue, negative correlations are in red.

**Table 1 ijerph-20-03183-t001:** General characteristics of study participants.

	Control (*n* = 21)	FMS (*n* = 19)	*p*-Value
Age	46.6 (8.7)	51.4 (7.4)	0.070
BMI	22.1 (2.7)	27.8 (4.8)	<0.001
Smoking status			0.043
Non-smokers	21 (100.0)	14 (3.7)	
Former smokers	0 (0)	1 (5.3)	
Current smokers	0 (0)	4 (21.1)	
Hypertension			0.085
Yes	20 (95.1)	14 (73.7)	
No	1 (4.8)	5 (26.3)	
Diabetes			1.000
Yes	19 (90.5)	17 (89.5)	
No	2 (9.5)	2 (10.5)	
Dyslipidemia			0.003
Yes	0 (0)	7 (36.8)	
No	21 (100)	12 (63.2)	
Total cholesterol	187.2 (39.1)	187.1 (34.1)	0.991
HDL-cholesterol	62.9 (15.4)	52.9 (12.8)	0.031
LDL-cholesterol	118.2 (35.9)	114.7 (17.3)	0.698
Triglyceride	75.5 (32.9)	182.2 (67.6)	<0.001
ESR	16.5 (13.9)	19.9 (15.2)	0.467
CRP	0.1 (0.2)	0.2 (0.3)	0.106
WPI		11.2 (5.2)	
Symptom severity scale		8.0 (2.6)	
VAS		5.9 (1.9)	
FIQ		61.5 (24.9)	

BMI, Body Mass Index; ESR, Erythrocyte Sedimentation Rate; CRP, C-Reactive Protein; WPI, Widespread pain index; VAS, Visual analogue scale; FIQ, Fibromyalgia Impact Questionnaire.

**Table 2 ijerph-20-03183-t002:** Difference in microbial abundance between FMS and the control at genus level.

	Coefficient	SE	W-Value	*p*-Value
*Eubacterium eligens group*	−3.67	0.84	−4.36	<0.001
*Eubacterium ruminantium group*	−1.41	0.74	−1.91	<0.001
*Alloprevotella*	−1.07	0.76	−1.41	<0.001
*Acinetobacter*	−0.49	0.38	−1.28	<0.001
*Rikenellaceae RC9 gut group*	−0.93	0.75	−1.25	<0.001
*Coprobacter*	−0.81	0.66	−1.23	<0.001
*Adlercreutzia*	−0.53	0.47	−1.15	<0.001
*Methanobrevibacter*	−0.36	0.39	−0.93	<0.001
*Anaerococcus*	−0.32	0.41	−0.78	<0.001
*Peptococcus*	−0.26	0.46	−0.56	<0.001
*Butyricicoccaceae*	−0.23	0.43	−0.54	<0.001
*Enterorhabdus*	−0.14	0.32	−0.44	<0.001
*Allisonella*	−0.19	0.45	−0.43	<0.001
*Gastranaerophilales*	−0.33	0.84	−0.39	<0.001
*Methylobacterium-Methylorubrum*	−0.13	0.33	−0.38	<0.001
*Granulicatella*	−0.15	0.44	−0.35	<0.001
*Terrisporobacter*	−0.18	0.54	−0.33	<0.001
*Gemella*	−0.15	0.46	−0.32	<0.001
*Finegoldia*	−0.11	0.43	−0.27	<0.001
*Campylobacter*	−0.09	0.39	−0.23	<0.001
*Parvimonas*	−0.07	0.38	−0.19	<0.001
*Prevotellaceae UCG-001*	−0.05	0.33	−0.16	<0.001
*Peptoniphilus*	0.01	0.40	0.03	<0.001
*Porphyromonas*	0.03	0.28	0.10	<0.001
*Ornithobacterium*	0.08	0.32	0.24	<0.001
*Corynebacterium*	0.12	0.47	0.25	<0.001
*Fenollaria*	0.10	0.32	0.31	<0.001
*Staphylococcus*	0.15	0.44	0.35	<0.001
*Actinomyces*	0.14	0.36	0.40	<0.001
*Fournierella*	0.23	0.45	0.50	<0.001
*Tuzzerella*	0.25	0.44	0.57	<0.001
*Faecalitalea*	0.42	0.67	0.63	<0.001
*Coprobacillus*	0.41	0.63	0.65	<0.001
*Succinivibrio*	0.48	0.71	0.67	<0.001
*Lachnospiraceae NC2004 group*	0.29	0.37	0.78	<0.001
*Slackia*	0.39	0.47	0.84	<0.001
*Victivallis*	0.43	0.48	0.89	<0.001
*Clostridia*	0.35	0.33	1.06	<0.001
*Eubacterium fissicatena group*	0.44	0.42	1.06	<0.001
*Howardella*	0.64	0.60	1.08	<0.001
*Megasphaera*	1.21	0.79	1.54	<0.001
*Paludicola*	1.30	0.50	2.60	<0.001
*Catenibacillus*	1.23	0.44	2.77	<0.001
*Eisenbergiella*	2.00	0.71	2.81	<0.001
*Caproiciproducens*	1.14	0.39	2.92	<0.001
*Frisingicoccus*	1.89	0.63	2.98	<0.001

SE, Standard error; A higher W-value indicates a more significant difference of abundance between two groups. A negative value indicates lower abundance than that in the control group, and a positive value indicates higher abundance than that in the control group.

## Data Availability

Data sharing not applicable. No new data were created or analyzed in this study. Data sharing is not applicable to this article.

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
