# Peer review of "Microbial Composition and Stool Short Chain Fatty Acid Levels in Fibromyalgia"

_ijerph, 2023, doi:10.3390/ijerph20043183_

Round 1

Reviewer 1 Report

Kim et al. investigated the microbial composition and stool SCFA levels in Korean women with FMS. Their findings complement the previous studies on the Caucasian FMS patient population. Overall, the manuscript is well-written, the methods are well-documented, and the results are nicely presented. 

Major comments:

  1. The figure legends need more information. For example, what is the y-axis in Figure 1? Does each dot represent one individual?
  2. How were the SCFA levels quantified in the fecal samples? The experimental details, like if standard curves were used, should be documented in the methods section. 

Minor comments:

  1. In line 325, the comment on pain sensation seems out of context since no experiments or observations were made on pain levels in the FMS groups vs. controls.
  2. Please expand section 3.5 to include the rationale for performing pairwise analysis and interpretation of the results.  
  3. There are typos in lines 63, 144, 162, and 334. 

Author Response

Dear Editor

Thank you for providing us with the opportunity to revise our manuscript titled Microbial composition and stool short-chain fatty acid levels in fibromyalgia. We have addressed all reviewers’ comments regarding the manuscript. We believe that our revisions have enhanced the quality of the manuscript and will provide readers with more comprehensive information regarding this research. The revised sections of the manuscript have been highlighted, and we have responded to the reviewers’ comments point-by-point, as shown below.

Reviewer 1

General comment

Kim et al. investigated the microbial composition and stool SCFA levels in Korean women with FMS. Their findings complement the previous studies on the Caucasian FMS patient population. Overall, the manuscript is well-written, the methods are well-documented, and the results are nicely presented.

[Answer:] Thank you for your valuable comments.

Major comments:

Comment #1

The figure legends need more information. For example, what is the y-axis in Figure 1? Does each dot represent one individual?

[Answer] Thank you for the comment. Dots indicate the diversity value of each fecal sample. We have modified the legend and added information on the y-axis in figure 1.

Comment #2

How were the SCFA levels quantified in the fecal samples? The experimental details, like if standard curves were used, should be documented in the methods section.

[Answer:] We agree with your comments. We have added the experimental details for the fecal SCFA analysis in the Methods section.

We revised the paragraph as follows :

“2.4 Measurement of SCFA in fecal samples

“SCFA were extracted from 0.2 g of fecal samples. Fecal samples were immediately frozen at -20 °C and transferred to a -70 °C freezer without preservatives. Fecal matter was first homogenized in three volumes of deionized water, centrifuged for 3 min at 13,000 rpm, and, finally, the supernatant was collected. The supernatant (150 uL) was placed in a 10-mL screw cap vial with 150 µL GC buffer solution. A solution containing (NH4)2SO4, NaH2PO4, and 2-ethylbutric acid was used as an internal standard. Stool SCFA were analyzed using a gas chromatography-mass spectrometry (GC-MS) system (7890B, Agilent Technologies, Santa Clara, CA, USA) equipped with a 7697A headspace sampler and flame ionization detector (FID) (Agilent Technologies). An HP-innowax capillary GC column (30 m x 0.32 mm x 0.25 μm; Agilent) was used with constant flow of nitrogen as a carrier gas. The operating conditions were as follows: oven temperature: 85 °C, loop temperature: 90 °C, transfer line temperature: 100 °C, and FID temperature: 250 °C. The column temperature was raised from 60 °C to 140 °C at 30 °C per minute, then increased to 170 °C at 30 °C per minute, and finally, to 180 °C at 40 °C per minute and held for 0.75 min. The homogeneity of the chromatographic peaks was verified using the extracted ions of characteristic fragments to optimize resolution and peak symmetry. Data analysis was performed using MassHunter WorkStation (Agilent Technologies). Concentrations of SCFA were expressed as μmol/g feces.”

Minor comments:

Comment #3

In line 325, the comment on pain sensation seems out of context since no experiments or observations were made on pain levels in the FMS groups vs. controls.

[Answer:] Thank you for your comments. We have deleted the comment regarding pain sensitization. Instead, we revised the text as follows:

“a potential association between FMS and microbital metabolites using stool metabolite analysis.”

Comment #4

Please expand section 3.5 to include the rationale for performing pairwise analysis and interpretation of the results. 

[Answer:] We agree with your comments. We have expanded the explanation of the heatmap analysis and the interpretation of the results. Revised section 3.5 and Figure 4 legends are as follows:

“3.5 Correlation between inflammatory markers, symptomatic scales of FMS and microbial diversity and SCFA

General correlations among the symptomatic scales of FMS, serum inflammatory markers as clinical variables, microbial diversity, and fecal SCFA levels are presented as Spearman's correlation heatmaps (Figure 4). In the subgroup analysis, symptomatic scales of FMS, such as the VAS, WPI, and FIQ, were not significantly correlated with the observed ASVs and Simpson and Shannon indices, which indicate microbial diversity. In addition, they were not significantly correlated with SCFA levels. Serum inflammatory markers were not significantly correlated with microbial diversity or fecal metabolites.

Figure 4. Heat map of Spearman correlation matrix for inflammatory markers, symptomatic scales of FMS, microbial diversity, and fecal metabolites. Color gradients indicate degree of correlation; positive correlations are marked in blue, negative correlations in red.”

Comment #5

There are typos in lines 63, 144, 162, and 334.

[Answer:] We revised the sentences as you mentioned.

Reviewer 2 Report

It shows the solidity of the work and the effort made by the authors of the manuscript.

I recommend that you use different keywords since they repeat the terms already used in the title. At the time of the search, they lose search power

In the introduction, there are parts that are more for the discussion than for an introduction. Authors must state the facts of the introduction.

Material and methods, it is true as stated in the limitations that the number of individuals is very small and this means that any change in one of the parameters will cause the statistics to be different.

The very fact of having more controls than cases is notable.

Table 1 shows the data or characteristics of the participants.

At this point, I would like the authors to clarify several things for me.

1.- how they valued that none of the control participants were smokers. This does not positively affect the microbiota. There were no smokers and they were excluded from the study.

2.- I am also curious about the fact that despite the fact that the controls have a lower BMI than the patients, however, all the controls have dyslipidemia. This data is correct.

3.- Regarding diabetes, I don't know if they have the data on whether they are type I or II

4.-Cholesterol values ​​are also striking in terms of their equality, while triacylglyceride values ​​seem clearer.

In the data of the table, they put a *. To which it refers

Table 2 where the genera of microbiota are described. However, it is not clear to me the differences between FMS and control.

3.5 could expand it a bit more

The conclusions obtained could be sufficient, even so, in the interviews one should also ask about the type of diet they follow.

Given that many are diabetics and even with hypertension, cholesterol, etc., knowing if the diet they follow influences their microbiota in any way. They comment that it is another limitation, but it would have been easily overcome if they had asked about eating habits in the interview.

Author Response

Dear Editor

Thank you for providing us with the opportunity to revise our manuscript titled Microbial composition and stool short-chain fatty acid levels in fibromyalgia. We have addressed all reviewers’ comments regarding the manuscript. We believe that our revisions have enhanced the quality of the manuscript and will provide readers with more comprehensive information regarding this research. The revised sections of the manuscript have been highlighted, and we have responded to the reviewers’ comments point-by-point, as shown below.

Reviewer 2

Comment #1

It shows the solidity of the work and the effort made by the authors of the manuscript.

I recommend that you use different keywords since they repeat the terms already used in the title. At the time of the search, they lose search power

[Answer:] We agree with your comments. We have added a few additional keywords as follows.

“Keywords: fibromyalgia; gastrointestinal microbiome; short chain fatty acids; metabolome; brain-gut axis”

Comment #2

In the introduction, there are parts that are more for the discussion than for an introduction. Authors must state the facts of the introduction.

Material and methods, it is true as stated in the limitations that the number of individuals is very small and this means that any change in one of the parameters will cause the statistics to be different. The very fact of having more controls than cases is notable.

[Answer:] Thank you for this valuable comment. We believe that the background information has been provided to enhance the clarity and readability of the manuscript because there are few studies evaluating microbial composition and short-chain fatty acids in FMS. However, if the reviewer guided us as to which part of the introduction should be removed or revised, we would be happy to revise the manuscript.

Comment #3

Table 1 shows the data or characteristics of the participants. At this point, I would like the authors to clarify several things for me. How they valued that none of the control participants were smokers. This does not positively affect the microbiota. There were no smokers and they were excluded from the study.

[Answer:] Thank you for valuable comments. We enrolled controls who underwent health checkups at the Department of Disease Prevention and Health Promotion; therefore, participants in the control group were unlikely to smoke. Smoking prevalence among Korean women is approximately 6.6 % based on the 2020 OECD health statistics1. On the basis of this factor, women who smoke may not have been included as our control group was also small. Additionally, we stated that the negative impact of smoking on the gut microbiome could not have been well reflected because there were no smokers.

Comment #4 I am also curious about the fact that despite the fact that the controls have a lower BMI than the patients, however, all the controls have dyslipidemia. This data is correct.

[Answer:] We apologize for the error we made in general characteristics of participants. The first and second rows of dyslipidemia should have been changed. We corrected this error in the tables.

Comment #5 Regarding diabetes, I don't know if they have the data on whether they are type I or II

 [Answer:] We agree with that the data on types of diabetes would have provided more information on the characteristics of study participants. Unfortunately, this information was not gathered at the time of data collection. This variable will be considered in our next study.

Comment #6 Cholesterol values ​​are also striking in terms of their equality, while triacylglyceride values ​​seem clearer.

[Answer:] Thank you for your thorough review. The cholesterol level of FMS and the control were 187.17 mg/dL and 187.24 mg/dL respectively. Despite the difference in triglyceride levels, we believe that the similar levels of cholesterol might be due to differences in the degree of variance in cholesterol levels between the two groups.

Comment #7 In the data of the table, they put a *. To which it refers

[Answer:] We apologize for the errors on the *. We removed this error from the table.

Comment #8 Table 2 where the genera of microbiota are described. However, it is not clear to me the differences between FMS and control.

[Answer:] We used ANCOM to compare the relative abundance of gut microbiota between two groups. ANCOM can approximate ASVs that are far from mean’s zero. If the differential abundance is 0, there is no significant difference between two groups; if 1, there is a significant difference. In table 2, genera with significant differences are described, where W is a statistical value. A higher W value indicates a more significant difference in abundance. A negative value indicates lower abundance than that in the control group, and a positive value indicates higher abundance than that in the control group. We have added this information in Table 2.

Comment #8 3.5 could expand it a bit more

  [Answer:] We revised explanation on correlation heatmap of 3.5 and Figure 4 legends.

Revised section 3.5 and Figure 4 legend are as below.

“3.5 Correlation between inflammatory markers, symptomatic scales of FMS and microbial diversity and SCFA

General correlations among the symptomatic scales of FMS, serum inflammatory markers as clinical variables, microbial diversity, and fecal SCFA levels are presented as Spearman's correlation heatmaps (Figure 4). In the subgroup analysis, symptomatic scales of FMS, such as the VAS, WPI, and FIQ, were not significantly correlated with the observed ASVs and Simpson and Shannon indices, which indicate microbial diversity. In addition, they were not significantly correlated with SCFA levels. Serum inflammatory markers were not significantly correlated with microbial diversity or fecal metabolites.

Figure 4. Heat map of Spearman correlation matrix for inflammatory markers, symptomatic scales of FMS, microbial diversity, and fecal metabolites. Color gradients indicate degree of correlation; positive correlations are marked in blue, negative correlations in red.”

Comment #9 The conclusions obtained could be sufficient, even so, in the interviews one should also ask about the type of diet they follow.

Given that many are diabetics and even with hypertension, cholesterol, etc., knowing if the diet they follow influences their microbiota in any way. They comment that it is another limitation, but it would have been easily overcome if they had asked about eating habits in the interview.

[Answer:] We agree with your valuable comments. The participants’ dietary habits should have been included because they could have influenced or interfered with the results. We have described this limitation in the Discussion section and intend to design future studies where we include dietary information.

Reference

  1. Organization for Economic Co-operation and Development. OECD Health Statistics 2020. Available at https://www.oecd.org

Round 2

Reviewer 2 Report

The changes introduced improve the manuscript as well as the explanations.